# Revisiting Cultural Issues in Suicide Rates: The Case of Western Countries

**DOI:** 10.3390/ijerph22040596

**Published:** 2025-04-10

**Authors:** Diego De Leo, Mujde Altin

**Affiliations:** 1Australian Institute for Suicide Research and Prevention, Griffith University, Nathan, QLD 4111, Australia; 2Slovene Centre for Suicide Research, Primorska University, 6000 Koper, Slovenia; 3De Leo Fund, 35137 Padova, Italy; mujde.altin@studenti.unipd.it

**Keywords:** suicide trends, cultural factors, Western countries, Latin countries, Anglo-Saxon countries

## Abstract

Suicidal behaviors among different age groups show epidemiological differences between countries. Specifically, suicide rates for the younger populations appear to be lower in Latin-origin countries (such as Italy, Spain, and Portugal) in comparison to other Western countries (especially Anglo-Saxon countries such as Canada, New Zealand, and Australia). The opposite seems to be true for the older population, suggesting a cross-cultural pattern for suicidal behavior in different ages. The current study replicates a study published in 1999 and compares suicide data between 1990 and 1994 with more recent data from the years 2016 and 2020 to investigate the persistence of previously observed trends. Basically, the recent years’ data confirm the patterns evidenced a quarter of a century ago, and substantially confirm the existence of suicide trends embedded with countries’ cultural factors and traditions. This investigation underlines the importance of incorporating anthropology, sociology, ethnography, and geography while studying culture-related patterns in suicide.

## 1. Introduction

Nearly every nation in the world is facing population aging. Both the overall number of individuals aged 65 years and over, and their proportion in society are increasing in percentage terms [1]. In 2020, the number of individuals aged 65 and above was around 727 million and it is estimated to reach 1.5 billion people by 2050—increasing from 9.3 percent to 16.0 percent [2]. For countries such as Italy, it is predicted that 35.9% of the population will be more than 65 years old within three decades [2,3].

When the distribution of suicidal behaviors across ages is examined, older ages constitute the greatest risk group globally [4,5]. Considering the impact of population aging, a proportioned increase in suicide rates of older adults potentially constitutes a serious public health problem. Although high suicide rates in late life can generally be seen around the world [6], significant differences between countries and different cultures are present as well, even among Western countries.

More than two decades ago, an epidemiological study [7] underlined the existence of marked cultural patterns in suicide rates related to age also among Western countries. In that study, the five-year-averaged suicide rates for 23 Western countries were calculated between 1990 and 1994, and observed among age groups (ten-year intervals). Latin countries, such as Portugal, Spain, and Italy, showed much lower suicide rates in younger individuals as compared to Western countries such as Australia, New Zealand, and Canada. Instead, lower rates of suicide were seen in older adults of the same Anglo-Saxon countries in comparison with the mentioned Latin countries (see Table 1). The trend lines expressing the age-related rates of those countries evidenced a progressive one approaching each other which became a real crossing over at the beginning of old age (65+). This pattern showed the same crossing over also in female subjects, suggesting the existence of a ‘cultural’ pattern in fatal suicidal behavior (see Table 2).

In this paper, we meant to verify whether that apparently culture-bound profile of suicide rates was not simply dependent on chance or historical period but it is still valid and representative of a ‘character’ of populations of different cultural backgrounds even if belonging to the same Western world. Thus, the main purpose of this study is to provide a comparative in-depth analysis of age-related suicide rates.

## 2. Methods

The suicide rates for 23 Western countries for the ages 1990–1994 were taken directly from De Leo’s work in 1999 (Table 1 and Table 2 prepared by Kerryn Neulinger, Australian Institute for Suicide Research and Prevention). For the recent data, rates were calculated between 2016 and 2020 for the same series of countries of the previous study. The data were taken from the World Health Organization’s (WHO) mortality database (https://platform.who.int/mortality/themes/theme-details/topics/indicator-groups/indicator-group-details/MDB/self-inflicted-injuries, accessed on 19 January 2025). Clusters of five-year rates for each country were then averaged. In case of missing data, the average was calculated on available years (for instance, for Portugal, rates for 2020 were not available, so the average rate was calculated for the years 2016–2019). In addition, the current WHO data cover broader age categories (compared to the data from the 1990s, calculated at 10-year intervals), and the tables are created accordingly. This does not pose a problem in terms of presenting general profiles and the purpose of the current study. Finally, all recent suicide rates were calculated separately for males and females, and can be found in Table 3 and Table 4.

## 3. Results

As displayed in Table 3, between 2016 and 2020, New Zealand, Australia, and Canada had the highest suicide rates for young males (15–24 years old), while Latin countries including Spain, Italy, and Portugal had the lowest ones. When we look at the older ages, we notice that the difference between these two Western groups decreases as a function of age. So to speak, the older subjects get, the smaller the gap for suicidality becomes. Particularly, while the suicide rates in Latin countries are on average 4–5 times lower than in Anglo-Saxon countries for the 15–24 age category, this rate drops to less than a 2-fold gap for the ages between 35 and 54. Interestingly, at 55–64, we start to observe a gradual crossing over between the Latin and Anglo-Saxon countries, which becomes increasingly visible for the 75-year-old and above category. For these age groups specifically, suicide rates are higher in Italy, Spain, and Portugal compared to the Anglo-Saxon group. The only exception in recent data is represented by older Australian men, whose rates stay slightly above Italy (24.2 vs. 22.9), but still below Portugal and Spain (47.2 and 30.5, respectively). Similarly to previous findings, the trend is present for the female subjects as well. As seen in Table 4, younger females show higher suicide rates in Canada, New Zealand, and Australia than in countries of Latin origin. On the contrary, older females’ suicide rates are higher for the Latin group, especially for Portugal and Spain (9.1 and 6.2, respectively). Similarly to data in males, Australia appears as an exception. However, this is possibly due to the fact that Italian rates for older males have halved, whilst Australian rates for the same age groups have reduced by a third. A similar reduction has been witnessed by Italian females, which halved their rates.

## 4. Discussion

Individuals who were born and have spent their youth in Latin countries appear at relatively low risk for suicidal behavior. In Anglo-Saxon countries, the opposite pattern occurs; in terms of suicidality, young individuals are in a more disadvantaged position compared to their peers in Italy, Spain, and Portugal. However, as they get older, they become less likely to die by suicide in comparison to older individuals who reside in Latin-origin countries.

Thus, the recent data we collected between 2016 and 2020 (displayed in Table 3 and Table 4) largely overlaps with those from 1990 to 1994 (Table 1 and Table 2), illustrating a permanent profile in suicidal rates. The culture-related cross-switch in age groups has remained pretty constant since the mid-90s, highlighting the importance of certain cultural and traditional values in the study of suicidal behavior. Despite the massive globalization that happened over the past years, the above-mentioned countries still preserve their initial profile in suicide.

As mentioned by Chandler et al. [8], suicidal behaviors are embodied and emplaced practices necessarily involving our bodies that are always socially, culturally, and materially located. Despite suicide being usually considered a complex phenomenon in the literature, the meanings of suicide in different cultures and its relationship with social practices have not been examined extensively [9]. Understanding suicide trends has mostly remained under the focus of mental health researchers and specifically ‘psy’ fields, such as psychology and psychiatry [9,10]. In fact, most of the existing studies focus on suicide independently of the cultural contexts and environments in which it occurs and acquires meaning [8,11]. As argued by Mills [12], certain contexts can be the cause of “hostile environment [s]… that make life, for some… unlivable and that incite, elicit, and invite suicidality” (p. 71).

The limited positioning (and medicalization) of suicide under the scope of ‘psy’ fields might be problematic in several ways. When observing mental disorders and psychological disturbances that are the main focus of these fields, we may notice clear differences regarding their prevalence around the world compared to suicidal behaviors. For instance, bipolar disorder and schizophrenia have a lifetime prevalence of circa 1% regardless of cultural contexts [13,14]. Some other psychological problems (such as depression) are relatively more culture-dependent, and their lifetime prevalence may vary across countries from 1% to 10% [15,16]. On the other hand, suicidal behaviors hold a dramatically different distribution around the world [14]. Geographic differences can be seen also within countries and among different racial/ethnic groups [17]. The possible reasons for this are indisputably complex and are the subject of ongoing research. One point that is clear, however, is that suicide cannot be fully understood without considering culture [18] or without being looked at through a ‘cultural’ lens [19]. 

Max Weber described culture as a “finite segment of the meaningless infinity of the world process” [20] (p. 37) and viewed humans as—somewhat active—agents who do produce culture and give it significance and meaning. On the other hand, anthropologist Clifford Geertz explained culture in terms of cultural learning and cultural symbols. In his view, culture can be seen as a cluster of “control mechanisms—plans, recipes, rules, instructions—what computer engineers call programs for the governing of behavior” [21] (p. 44). Through the cultural systems people, consciously and unconsciously, internalize and integrate established meanings and symbols to guide their behaviors and perceptions [22].The purpose of drawing attention to this point is not to neglect the dynamic, changeable, flexible, and diverse nature of culture, but rather to reveal the relationship between the stability we observe in some behavioral patterns and some cultural values and judgments established within it. The epidemiological data we have presented previously on suicide rates remind us of the utmost necessity of understanding socio-cultural factors while studying suicide [23,24].

The study has several shortcomings. Since the processing and official publication of suicide data varies from country to country, we used the findings from 2016 to 2020, the most up-to-date data available for comparative analysis. In order not to deviate from the aim of the study, we focused on two main Western groups. Future research should examine culture-related suicide patterns in other countries too, especially in the non-Western world.

Finally, rather than charting the overall year-to-year variation in suicide rates, the current work focused on comparing age-related high-risk groups in suicide rates. The following studies should address in more detail the various cultural factors (e.g., social norms, economic structure, ageism, and suicide policies) that contribute to these patterns.

## 5. Conclusions

‘Culture’ can be operationalized as an all-embracing term that defines the relationship of individuals to their environments. In this way, the study of this type of influence should not be dismissed when trying to understand and interpret suicidal behaviors. The current study provided some support for this perspective. Considering the distribution of age-related suicide rates across different Western countries, this investigation compared 1990–1994 and 2016–2020 suicide data, and revealed that age-related suicide profiles have not changed despite the rapidly changing world and the ongoing process of globalization, showing consistent patterns. This suggests that an important future direction in suicidology should include holding a multi-disciplinary perspective, with a particular inclusion and integration of anthropology, sociology, ethnography, critical cultural studies, and geography.

## Figures and Tables

**Table 1 ijerph-22-00596-t001:** Rank ordering of mean male suicide rates for 23 Western countries 1990–1994.

By 10-Year Age Groups, 1990–1994							
Rank	15–24	25–34	35–44	45–54	55–64	65–74	75+
1st	Finland 41.4	Finland 60.7	Hungary 82.0	Hungary 95.1	Hungary 84.6	Hungary 92.5	Hungary 183.0
2nd	**N. Zealand 39.0**	Hungary 54.4	Finland 67.8	Finland 64.1	Finland 57.3	Austria 61.1	Austria 118.0
3rd	Switzerland 25.8	Switzerland 32.7	France 40.1	Denmark 47.5	Austria 46.7	Belgium 50.4	France 103.0
4th	**Australia 25.7**	**N. Zealand 32.0**	Denmark 38.1	Austria 41.5	Denmark 42.6	Switzerland 47.4	Belgium 98.6
5th	**Canada 25.2**	France 32.0	Austria 37.2	France 40.1	Switzerland 41.9	France 47.1	Switzerland 89.8
6th	Norway 24.9	Belgium 30.5	Belgium 35.6	Switzerland 39.8	Belgium 38.9	Denmark 46.4	Germany 86.1
7th	Austria 24.3	Austria 30.3	Switzerland 33.0	Belgium 36.2	France 38.1	Finland 45.9	Denmark 74.9
8th	USA 21.9	**Australia 29.0**	Sweden 29.3	Sweden 31.9	Germany 32.2	Germany 35.9	Finland 71.9
9th	Hungary 20.1	**Canada 29.0**	**Canada 27.3**	Germany 31.1	Sweden 30.7	Sweden 33.7	** *Portugal 59.1* **
10th	Scotland 19.0	Ireland 27.1	Norway 26.9	Norway 28.8	Norway 28.8	USA 30.9	USA 55.4
11th	Ireland 18.3	Denmark 26.4	Scotland 26.2	**Canada 25.6**	Ireland 25.9	Norway 30.7	Sweden 51.9
12th	N. Ireland 17.6	Norway 26.1	Germany 26.0	Scotland 24.2	USA 25.0	** *Portugal 30.1* **	** *Spain 47.8* **
13th	Belgium 15.7	Scotland 26.1	**Australia 25.2**	**Australia 24.2**	**Canada 24.2**	**Australia 24.4**	** *Italy 44.3* **
14th	France 15.3	USA 24.6	**N. Zealand 23.9**	**N. Zealand 24.2**	**N. Zealand 23.2**	** *Spain 23.2* **	Netherlands 35.4
15th	Germany 14.0	Sweden 23.9	USA 23.5	USA 23.1	**Australia 22.9**	** *Italy 22.9* **	**Australia 32.8**
16th	Sweden 13.4	N. Ireland 22.4	Ireland 22.9	Ireland 19.6	** *Portugal 21.5* **	**Canada 22.1**	Norway 31.8
17th	Denmark 13.0	Germany 21.3	Netherlands 17.7	Netherlands 16.7	Netherlands 18.6	**N. Zealand 21.2**	**N. Zealand 29.8**
18th	England 11.1	England 16.3	England 17.4	England 16.2	Scotland 18.1	Netherlands 19.7	**Canada 28.9**
19th	Netherlands 9.3	Netherlands 15.9	N. Ireland 15.5	N. Ireland 15.1	N. Ireland 17.4	Ireland 18.3	England 17.1
20th	** *Spain 7.0* **	** *Portugal 13.2* **	** *Portugal 11.8* **	** *Portugal 14.6* **	** *Spain 17.4* **	Scotland 14.3	Scotland 16.0
21st	** *Italy 6.1* **	** *Spain 10.6* **	** *Italy 10.6* **	** *Italy 12.6* **	** *Italy 17.1* **	N. Ireland 12.8	Greece 15.8
22nd	** *Portugal 5.8* **	** *Italy 10.3* **	** *Spain 9.4* **	** *Spain 11.9* **	England 12.8	England 11.9	Ireland 13.8
23rd	Greece 4.0	Greece 5.6	Greece 5.9	Greece 6.7	Greece 7.8	Greece 10.1	N. Ireland 13.3

**Table 2 ijerph-22-00596-t002:** Rank ordering of mean female suicide rates for 23 Western countries 1990–1994.

By 10-Year Age Groups, 1990–1994							
Rank	15–24	25–34	35–44	45–54	55–64	65–74	75+
1st	Finland 7.5	Finland 12.0	Hungary 20.3	Hungary 26.5	Denmark 28.5	Hungary 37.6	Hungary 67.3
2nd	Austria 6.2	Belgium 11.8	Finland 17.4	Denmark 25.5	Hungary 28.0	Denmark 31.5	Denmark 30.2
3rd	Hungary 6.2	Hungary 11.6	Denmark 15.7	Finland 20.4	Belgium 17.9	Belgium 23.5	Austria 28.5
4th	**N. Zealand 6.2**	Sweden 10.1	Belgium 14.5	Belgium 18.2	France 17.6	Switzerland 19.8	Germany 26.4
5th	Sweden 5.9	Switzerland 9.0	Switzerland 13.5	Austria 17.1	Finland 17.5	Austria 18.5	France 25.3
6th	Switzerland 5.8	France 9.0	France 13.0	Switzerland 16.7	Austria 17.4	France 17.9	Belgium 24.2
7th	Norway 5.5	Scotland 8.3	Austria 12.1	France 16.5	Switzerland 17.0	Germany 16.7	Switzerland 23.0
8th	**Australia 5.1**	Austria 8.0	Sweden 11.8	Sweden 15.0	Sweden 15.4	Sweden 13.5	Sweden 14.2
9th	Belgium 5.1	Denmark 7.7	Norway 9.8	Germany 12.1	Germany 12.9	Finland 13.3	** *Portugal 12.2* **
10th	**Canada 4.9**	**N. Zealand 7.3**	Netherlands 9.6	Norway 11.5	Norway 12.0	Norway 12.4	Netherlands 12.1
11th	France 4.5	Netherlands 7.2	**Canada 8.1**	Netherlands 9.5	Netherlands 10.9	Netherlands 10.4	** *Spain 11.9* **
12th	USA 3.8	Norway 7.1	Germany 7.7	N. Ireland 9.4	**N. Zealand 7.7**	** *Spain 8.8* **	Finland 9.6
13th	Netherlands 3.7	Ireland 6.7	**N. Zealand 6.9**	**N. Zealand 8.9**	Ireland 7.7	** *Portugal 81.* **	** *Italy 9.3* **
14th	Scotland 3.7	**Australia 6.6**	Scotland 6.8	**Canada 8.1**	**Australia 6.9**	** *Italy 8.0* **	Norway 9.2
15th	Germany 3.5	**Canada 6.4**	N. Ireland 6.8	USA 7.3	USA 6.8	**N. Zealand 6.6**	**Australia 8.0**
16th	Denmark 3.3	Germany 5.7	**Australia 6.6**	Scotland 7.2	** *Italy 6.8* **	**Australia 6.6**	Scotland 6.0
17th	Ireland 2.5	USA 5.3	USA 6.6	**Australia 7.0**	Scotland 6.7	Ireland 6.4	USA 6.0
18th	N. Ireland 2.4	N. Ireland 3.9	Ireland 4.8	Ireland 6.8	**Canada 6.4**	Scotland 6.4	England 5.9
19th	** *Portugal 2.2* **	** *Portugal 3.5* **	England 3.9	** *Portugal 5.0* **	** *Portugal 6.2* **	USA 6.2	**Canada 4.7**
20th	England 2.1	England 3.5	** *Italy 3.9* **	** *Italy 4.9* **	** *Spain 6.0* **	**Canada 6.1**	**N. Zealand 4.3**
21st	** *Italy 1.8* **	** *Italy 2.9* **	** *Portugal 3.8* **	England 4.7	N. Ireland 4.8	England 5.2	Greece 3.4
22nd	** *Spain 1.7* **	** *Spain 2.6* **	** *Spain 3.0* **	** *Spain 3.9* **	England 4.7	N. Ireland 3.9	Ireland 3.0
23rd	Greece 0.7	Greece 1.4	Greece 1.3	Greece 2.3	Greece 2.4	Greece 2.8	N. Ireland 2.5

**Table 3 ijerph-22-00596-t003:** Rank ordering of mean male suicide rates for 23 Western countries.

By 10-Year Age Groups, 2016–2020					
Rank	15–24	25–34	35–54	55–74	75+
1st	**N. Zealand 24.9**	Finland 29.6	Belgium 31.5	Hungary 44.8	Hungary 87.8
2nd	USA 21.7	**N. Zealand 27.7**	USA 28.2	Belgium 30.0	Austria 72.4
3rd	Finland 20.2	USA 27.5	France 28.0	Austria 29.6	France 53.6
4th	**Australia 20.2**	**Australia 24.6**	Hungary 27.6	USA 28.2	Switzerland 48.5
5th	**Canada 16.8**	Norway 22.3	**Australia 27.4**	France 27.5	** *Portugal 47.2* **
6th	Sweden 14.3	Belgium 21.0	Finland 25.9	Switzerland 25.2	Germany 47.1
7th	Ireland 13.0	**Canada 20.0**	**Canada 23.0**	Finland 24.5	Belgium 43.2
8th	Switzerland 11.9	Sweden 18.9	Norway 22.3	Sweden 22.7	USA 39.9
9th	Norway 11.9	Ireland 17.7	**N. Zealand 22.0**	Germany 22.6	Denmark 36.9
10th	Belgium 11.6	Hungary 17.3	Ireland 21.9	**Australia 21.5**	Finland 32.2
11th	Austria 10.5	France 15.7	Netherlands 20.1	Denmark 21.4	** *Spain 30.5* **
12th	UK 10.0	UK 15.7	Austria 19.4	** *Portugal 21.3* **	Sweden 29.2
13th	Hungary 9.7	Netherlands 13.7	Sweden 19.4	**Canada 20.9**	**Australia 24.2**
14th	Netherlands 8.6	Austria 12.9	UK 19.3	Netherlands 19.6	** *Italy 22.0* **
15th	Germany 8.1	Switzerland 12.7	Denmark 18.2	Norway 18.0	**N. Zealand 21.7**
16th	Denmark 7.6	Germany 11.9	Switzerland 17.4	**N. Zealand 17.7**	**Canada 20.9**
17th	France 7.1	Denmark 11.1	Germany 17.4	Ireland 17.4	Netherlands 20.4
18th	** *Italy 5.1* **	** *Portugal 9.9* **	** *Portugal 15.9* **	** *Spain 15.8* **	Norway 20.0
19th	** *Spain 4.8* **	** *Spain 8.7* **	** *Spain 13.9* **	** *Italy 13.2* **	Greece 15.1
20th	** *Portugal 4.8* **	** *Italy 7.8* **	** *Italy 11.0* **	UK 12.8	UK 11.6
21st	Greece 4.1	Greece 6.5	Greece 8.9	Greece 11.6	Ireland 7.7

Note: Countries marked in bold letters show significant trends in accordance with the main subject of the study.

**Table 4 ijerph-22-00596-t004:** Rank ordering of mean female suicide rates for 23 Western countries.

By 10-Year Age Groups, 2016–2020					
Rank	15–24	25–34	35–54	55–64	75+
1st	Finland 10.5	**N. Zealand 9.0**	Belgium 11.8	Belgium 13.2	Hungary 18.6
2nd	Norway 9.0	Sweden 8.5	Norway 10.5	Hungary 11.7	Austria 12.3
3rd	**N. Zealand 8.1**	Finland 7.7	USA 9.2	Netherlands 10.4	France 11.0
4th	Sweden 7.4	**Australia 7.6**	Sweden 9.0	Norway 10.1	Denmark 10.9
5th	**Canada 7.3**	Norway 7.4	**N. Zealand 8.8**	France 9.3	Germany 10.8
6th	**Australia 6.9**	USA 7.0	**Australia 8.8**	Switzerland 9.2	Belgium 10.8
7th	Ireland 5.6	**Canada 6.5**	Netherlands 8.6	Sweden 9.1	Switzerland 10.1
8th	USA 5.5	Netherlands 6.3	Finland 8.3	Denmark 8.2	** *Portugal 9.1* **
9th	Netherlands 5.0	Belgium 6.0	France 8.0	Austria 8.1	Netherlands 8.5
10th	Switzerland 4.7	Ireland 4.6	Hungary 7.6	USA 7.8	Sweden 8.2
11th	Belgium 4.1	Switzerland 4.5	**Canada 7.5**	Finland 7.6	**Australia 6.9**
12th	UK 3.7	UK 4.3	Switzerland 7.3	Germany 7.3	** *Spain 6.2* **
13th	Denmark 3.6	France 4.1	Austria 6.3	** *Portugal 6.9* **	Finland 5.6
14th	Hungary 3.2	Denmark 3.6	Denmark 5.9	**Australia 6.6**	Norway 5.1
15th	Austria 2.8	Hungary 3.5	Ireland 5.8	**Canada 6.3**	** *Italy 4.1* **
16th	Germany 2.8	Germany 3.3	Germany 5.7	** *Spain 5.9* **	USA 4.1
17th	France 2.7	Austria 3.2	UK 5.5	**N. Zealand 5.1**	**Canada 4.0**
18th	** *Spain 1.7* **	** *Spain 2.4* **	** *Portugal 4.7* **	Ireland 4.1	**N. Zealand 3.6**
19th	** *Portugal 1.6* **	** *Portugal 2.4* **	** *Spain 4.6* **	UK 4.0	UK 2.9
20th	Greece 1.5	** *Italy 2.0* **	** *Italy 3.2* **	** *Italy 3.5* **	Greece 2.3
21st	** *Italy 1.3* **	Greece 1.6	Greece 2.3	Greece 2.0	Ireland 1.9

Note: Countries marked in bold letters show significant trends in accordance with the main subject of the study.

## Data Availability

The raw data supporting the conclusions of this article will be made available by the authors on request.

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
