# Peer review of "Revisiting Cultural Issues in Suicide Rates: The Case of Western Countries"

_ijerph, 2025, doi:10.3390/ijerph22040596_

Round 1
Reviewer 1 Report
Comments and Suggestions for Authors
Dear authors,
I congratulate you on their work. I make considerations intending to qualify the study:
- From a scientific writing point of view, the introduction section is confused with the results section and needs to be improved. What is referred to in “Table 1” and “Table 2” in the introduction are data reported in the study cited as “De Leo, 1999” by the authors. Despite this, the tables cited in the introduction are presented as results of the present study. I suggest keeping only the citation of the original study and reviewing/removing these links to the results tables, taking into account that the introduction should basically provide contextualization, presentation of the state of the art and pointing out the gap in the object of study (which could even be improved in the writing of this study).
- https://doi.org/10.1038/s43587-021-00160-1 (reference number 7) leads to an article different from the one described (De Leo, D. Late-life suicide in an aging world. Nat Aging 2, 7–12, 2022). Need to correct.
- I suggest writing the objective and hypothesis of the study more clearly, as they can be confusing.
- As tables 1 and 2 were used as a way to provide comparative data for suicide rates/age for the 23 countries between 1990-1994, they should be redirected to the end of the methods section (suggestion).
- Presentation of results is clear.
- Discussion needs to return to the study hypothesis and describe its limits.
- Conclusion is clear and relevant.
- References are adequate, and as far as possible, current.
Best regards.
Author Response
Reviewer 1
- From a scientific writing point of view, the introduction section is confused with the results section and needs to be improved. What is referred to in “Table 1” and “Table 2” in the introduction are data reported in the study cited as “De Leo, 1999” by the authors. Despite this, the tables cited in the introduction are presented as results of the present study. I suggest keeping only the citation of the original study and reviewing/removing these links to the results tables, taking into account that the introduction should basically provide contextualization, presentation of the state of the art and pointing out the gap in the object of study (which could even be improved in the writing of this study).
RESPONSE: We moved Table 1 & 2 at the end of the methods section following point 4, as a way to provide comparative data for suicide rates/age for the 23 countries between 1990-1994. The results section instead includes only the recent findings we calculated (Table 3 & 4).
TEXT CHANGE: page 2 & 3 (methods section)
- https://doi.org/10.1038/s43587-021-00160-1 (reference number 7) leads to an article different from the one described (De Leo, D. Late-life suicide in an aging world. Nat Aging 2, 7–12, 2022). Need to correct.
RESPONSE: We updated the reference in text and bibliography.
TEXT CHANGE: page 1 (paragraph 2) and reference number 7.
- I suggest writing the objective and hypothesis of the study more clearly, as they can be confusing.
RESPONSE: we modified the last paragraph of the introduction section to clarify the goal of the study.
TEXT CHANGE: page 2, paragraph 2
“In this paper we meant to verify whether that apparently culture-bound profile of suicide rates was not simply dependent on chance or historical period, but it is still valid and representative of a ‘character’ of populations of different cultural backgrounds even if belonging to the same western world. Thus, the main purpose of this study is to provide a comparative in-depth analysis of age-related suicide rates.”
- As tables 1 and 2 were used as a way to provide comparative data for suicide rates/age for the 23 countries between 1990-1994, they should be redirected to the end of the methods section (suggestion).
RESPONSE: We moved tables 1 & 2 to the end of the methods section.
- Presentation of results is clear.
- Discussion needs to return to the study hypothesis and describe its limits.
RESPONSE: The research we conducted is an in-depth exploratory study rather than a hypothesis-driven analysis. We added a new paragraph to the discussion section highlighting the limitations.
TEXT CHANGE: page 5, last paragraph
“The study has several shortcomings. Since the processing and official publication of suicide data varies from country to country, we used the findings from 2016-2020, the most up-to-date data available for comparative analysis. In order not to deviate from the aim of the study, we focused on two main Western groups. Future research should examine culture related suicide patterns in other countries too, especially in the non-Western world.”
- Conclusion is clear and relevant.
- References are adequate, and as far as possible, current.
Reviewer 2 Report
Comments and Suggestions for Authors
This revised manuscript presents results of the compilation of suicide rates for over 20 nations in Western countries for 2016-2020, following up on an earlier study of many of the same nations from 1990-1994. The authors focus on culture as a factor in suicide.
Strengths of the manuscript include the presentation of suicide levels for males and females in a variety of Western nations, providing the rates by both sex and age groupings, deriving the data from the primary source of international mortality data (i.e., WHO), averaging the figures over a multiyear period to lessen the annual variations that are common in such data, and the focus on culture as an influence on suicide (a factor that is often not emphasized).
There are possible minor issues that might be addressed.
- Though beyond the scope and conceptual focus of the manuscript, there are likely some nations in which the primary higher risk sex-age groupings have remained the same for these two periods, but there have been shifts in some nations’ rankings among the tabled data of the two time periods. For instance, the USA maintains the same relative high risk groups for the two periods but the relative ranking among the nations presented are higher at nearly every sex-age ranking for 2016-2020 compared to that of 1990-1994. USA suicide rates have risen to their highest levels in 80 years from the lower rates of the end of the 1990s. There are likely other nations among those provided that have had parallel changes (either upward or downward). Perhaps a recognition that changes in rate levels are not the focus of the study, but rather the relative high risk groupings that seem to be “cultural” in nature might be included. No specification of the kinds of specific cultural practices and characteristics that the authors are emphasizing are provided; an example of some such forces within even one of the nations included might be informative.
- The data in the tables are informative, but I found no reason provided for the nations that appear in bold type in the tables compared to the others that do not. Some “note” with the tables might provide an explanation for the differences in presentation. The text does seem to highlight these particular nations in its discussion and introduction, so perhaps that is the explanation, but some clarification of this would seem desirable.
- The potentially important inclusion of culture to understand suicide rates and their diversity is certainly worthy of emphasis, but it might also be recognized that the changes in rates for the two time periods are likely influenced particularly by the historical events and the passage of time as well. The major forces occurring as a result of economic, political, and other national and international social events (financial crises, conflicts/wars, famines, droughts and other weather and climatic changes, assassinations, diseases and pandemics such as COVID-19, etc.) that occur over time within and across nations may well account for some and in some cases many of these nations after over 20 years. This issue might be acknowledged as another force in suicide within nations – a further demonstration of the complexity the authors acknowledge. These forces are also shapers of “culture.”
Author Response
Reviewer 2
There are possible minor issues that might be addressed.
- Though beyond the scope and conceptual focus of the manuscript, there are likely some nations in which the primary higher risk sex-age groupings have remained the same for these two periods, but there have been shifts in some nations’ rankings among the tabled data of the two time periods. For instance, the USA maintains the same relative high risk groups for the two periods but the relative ranking among the nations presented are higher at nearly every sex-age ranking for 2016-2020 compared to that of 1990-1994. USA suicide rates have risen to their highest levels in 80 years from the lower rates of the end of the 1990s. There are likely other nations among those provided that have had parallel changes (either upward or downward). Perhaps a recognition that changes in rate levels are not the focus of the study, but rather the relative high risk groupings that seem to be “cultural” in nature might be included. No specification of the kinds of specific cultural practices and characteristics that the authors are emphasizing are provided; an example of some such forces within even one of the nations included might be informative.
RESPONSE: we added a new paragraph to discussion.
TEXT CHANGE: page 6, paragraph 1
“Finally, rather than charting the overall year-to-year variation in suicide rates, the current work focused on comparing age-related high-risk groups in suicide rates. Following studies should address in more detail the various cultural factors (e.g., social norms, economic structure, ageism, suicide policies) that contribute to these patterns.”
- The data in the tables are informative, but I found no reason provided for the nations that appear in bold type in the tables compared to the others that do not. Some “note” with the tables might provide an explanation for the differences in presentation. The text does seem to highlight these particular nations in its discussion and introduction, so perhaps that is the explanation, but some clarification of this would seem desirable.
RESPONSE: we added a clarification note under Table 3 & 4 in the results section.
TEXT CHANGE: page 4
“Note: Countries marked in bold letters show significant trends in accordance with the main subject of the study.”
- The potentially important inclusion of culture to understand suicide rates and their diversity is certainly worthy of emphasis, but it might also be recognized that the changes in rates for the two time periods are likely influenced particularly by the historical events and the passage of time as well. The major forces occurring as a result of economic, political, and other national and international social events (financial crises, conflicts/wars, famines, droughts and other weather and climatic changes, assassinations, diseases and pandemics such as COVID-19, etc.) that occur over time within and across nations may well account for some and in some cases many of these nations after over 20 years. This issue might be acknowledged as another force in suicide within nations – a further demonstration of the complexity the authors acknowledge. These forces are also shapers of “culture.”
RESPONSE: we acknowledge that the changes in rates for the two time periods are clearly related to historical events and subsequent social transitions, viewing these forces as fundamental components of cultural change. In the discussion section, we mention how most of the research in “psy” fields ignores the cultural context or environment in which suicide occurs and acquires meaning. In fact, evaluating the environment (in the context of space and time) certainly requires taking into account factors such as financial crises, economic shifts, pandemics etc.